# Insomnia and Personality—A Network Approach

**DOI:** 10.3390/brainsci7030028

**Published:** 2017-03-02

**Authors:** Kim Dekker, Tessa F. Blanken, Eus J. W. Van Someren

**Affiliations:** 1Department of Sleep and Cognition, Netherlands Institute for Neuroscience, Meibergdreef 47, Amsterdam 1105 BA, The Netherlands; t.blanken@nin.knaw.nl (T.F.B.); eusvansomeren@gmail.com (E.J.W.V.S.); 2Department of Integrative Neurophysiology and Psychiatry, Center for Neurogenomics and Cognitive Research (CNCR), Neuroscience Campus Amsterdam, VU University and Medical Center, A.J. Ernststraat 1187, Amsterdam 1081 HL, The Netherlands

**Keywords:** network analysis, insomnia, five factor model personality traits, daytime complaints, neuroticism

## Abstract

Studies on personality traits and insomnia have remained inconclusive about which traits show the most direct associations with insomnia severity. It has moreover hardly been explored how traits relate to specific characteristics of insomnia. We here used network analysis in a large sample (*N* = 2089) to obtain an integrated view on the associations of personality traits with both overall insomnia severity and different insomnia characteristics, while distinguishing direct from indirect associations. We first estimated a network describing the associations among the five factor model personality traits and overall insomnia severity. Overall insomnia severity was associated with neuroticism, agreeableness, and openness. Subsequently, we estimated a separate network describing the associations among the personality traits and each of the seven individual items of the Insomnia Severity Index. This revealed relatively separate clusters of daytime and nocturnal insomnia complaints, that both contributed to dissatisfaction with sleep, and were both most directly associated with neuroticism and conscientiousness. The approach revealed the strongest direct associations between personality traits and the severity of different insomnia characteristics and overall insomnia severity. Differentiating them from indirect associations identified the targets for improving Cognitive Behavioral Therapy for insomnia with the highest probability of effectively changing the network of associated complaints.

## 1. Introduction

Insomnia is a common burden in the general population [1,2]. Insomnia disorder can be diagnosed if subjective problems with initiating sleep, maintaining sleep, or waking up too early occur at least three nights a week, persist for at least three months, and are accompanied by at least one form of subjective daytime impairments like fatigue, malaise, or difficulties with concentration [3]. The diagnosis of insomnia disorder thus requires the presence of both nocturnal and daytime complaints. Although the causes of insomnia are still poorly understood [4], a prevailing theory by Spielman et al. suggests the involvement of three types of factors [5]: premorbid predispositions, precipitating factors, and perpetuating factors. It has been suggested that certain personality traits may predispose to insomnia [6,7,8,9,10].

The dominant model of personality is the five-factor model (FFM), or the Big Five, and is based on a substantive body of evidence that found a five-factor solution to the correlations among personality characteristics [11]. The five personality traits are extraversion, agreeableness, conscientiousness, neuroticism, and openness. The study of personality traits in insomnia may provide clues about the underlying brain circuits involved in insomnia, because individual differences in personality traits are associated with individual differences in brain structure and brain function [12,13,14]. Some of the reported associations indeed seem relevant to insomnia, for example, both sleep vulnerability and introversion have been linked to low gray matter density in the orbitofrontal cortex [15,16,17].

Even though the association between personality traits and insomnia has been studied extensively [7,8,9,18,19,20,21,22,23,24,25,26,27], there is no conclusive consistency about which of the personality traits are most strongly associated with insomnia in a general adult population. Within the framework of the prevailing five-factor model (FFM) of personality [28], the most consistently reported finding in insomnia disorder is high neuroticism [7,8,18,20,22,25,26]. Several studies also suggested insomnia disorder to be associated with low conscientiousness, but showed that when all personality traits were simultaneously evaluated in a single regression model [18,19,20,22], this association appeared to be secondary to the inverse association of conscientiousness with neuroticism. Multi-collinearity of personality traits thus has to be considered in efforts to understand the complexity of direct and indirect relations between insomnia and personality traits.

A relatively unexplored aspect is whether specific symptoms of insomnia may be associated differentially with personality traits. Previous studies focused mostly on nocturnal symptoms of insomnia [18,19,20], or on a compound score [19,22]. Since insomnia is defined by both nocturnal and daytime aspects, it may be that some personality traits relate to nocturnal complaints, whereas others relate to daytime complaints. It could be more informative to perform an integrated analysis on how different symptoms of insomnia are associated with different personality traits.

Such an integrated analysis of how traits and symptoms are associated has shown to be feasible using network analysis. This method can simultaneously estimate partial correlations between all variables included—in our case all personality traits and insomnia symptoms measured—and visualizes them in a so-called concentration network graph. The graph shows all variables as nodes, and their partial correlations as connecting edges [29]. The relative strength of partial correlations can be represented by the length, color saturation, and width of the edges between each of the nodes. Network analysis has recently been introduced for psychometric data [30], including personality traits [29,31], and has found increasing popularity over the years [32].

The present study is, as far as we know, the first to apply network analyses on the data of a large sample of volunteers to obtain an integrated view on the associations of personality traits with overall insomnia severity as well as with different symptoms of insomnia. We estimated and visualized two concentration networks: one including the five personality traits and the Insomnia Severity Index (ISI) summary score; the other including the five personality traits and each of the seven individual ISI items.

## 2. Materials and Methods

### 2.1. Participants

The data were obtained through the Netherlands Sleep Registry (NSR) [33]. The NSR is an internet platform that aims to asses a wide variety of traits across the general population, both good and bad sleepers [33], in order to create a psychometric database to facilitate research on traits that distinguish insomniacs from good sleepers. Participants of the NSR are unpaid volunteers that anonymously fill out as many different questionnaires as they wish, at a self-chosen place and time. Participants fill out each of the questionnaires once. Commitment is supported by online feedback, newsletters, and occasional voucher lotteries. For the present study, we included all participants that completed both the Insomnia Severity Index (ISI) [34], the Mini-International Personality Item Pool (Mini-IPIP) [35], and a questionnaire on demographics including age and sex. Participants younger than 18 were excluded. This resulted in a cross-sectional sample of 2089 volunteers (1573 females, 75.3%), with an age range of 18 to 84 years of age (Mean age 51.7 ± 13.6). The questionnaires were filled out between May 2012 and October 2016. The median time difference between filling out ISI and Mini-IPIP was 7.4 months. The Medical Ethical Committee of the Academic Medical Center of the University of Amsterdam (29 September 2009, 09.17.1396) as well as the Central Committee on Research Involving Human Subjects (CCMO, 14 December 2011, CCMO11.1813/GK/jt), The Hague, the Netherlands, approved of unsigned informed consent, because of the voluntary and anonymous nature of participation and lack of any intervention or behavioral constraint.

### 2.2. Materials

Mini-IPIP—The 20-item Mini-IPIP [35] assesses the Big Five personality traits of neuroticism, conscientiousness, agreeableness, extraversion, and openness to experiences. Participants are asked to rate how accurate brief statements describe themselves, using a five-point Likert scale ranging from 1 (very inaccurate) to 5 (very accurate).

ISI—The Insomnia Severity Scale (ISI) is a seven-item scale that addresses aspects of insomnia [34]. Each question uses a five-point Likert scale, ranging from 0 to 4. The first three questions assess the three main sleep complaints: difficulty initiating sleep (DIS), difficulty maintaining sleep (DMS) and early morning awakening (EMA). Questions 4–7 assess respectively dissatisfaction with sleep (Dissat), interference with daily functioning (IDF), how noticeable sleep problems impair quality of life (NIQoL), and worry about sleep (Worry). The compound score is a simple sum score of all items, resulting in a score ranging from 0 to 28.

### 2.3. Statistical Analysis

To estimate and visualize the concentration network of associations between the five personality traits and the Insomnia Severity Index (ISI) summary score, partial correlations were calculated between each of the six variables. The partial correlation among two variables represents the strength of the direct association between these variables, while taking all the other variables into account. The networks of the partial correlations thus represent the unique association between any two variables, that cannot be explained by their common associations with other variables [29]. As such, the network structure highlights possible pathways of influence: unrelated variables cannot directly influence each other, while the presence of a direct association indicates a potential causal pathway between two variables. To minimize spurious associations due to sampling error, we applied Least Absolute Shrinkage and Selection Operator (LASSO) regularization. This procedure controls for false-positive associations and retrieves only the most robust associations. Therefore, the included associations are very likely to play an important role in the network architecture (see Epskamp and Fried [36] for more information).

To visualize the concentration networks, each of the six variables is shown as a node, connected by edges. The edges in the network represent the partial correlations between variables that were estimated by the network model using LASSO regularization [29]. Green edges correspond to positive partial correlations, while red edges correspond to negative partial correlations. In addition, both the edge width and color saturation are scaled to the strength of the association: wider edges and more saturated colors represent stronger associations. The Fruchterman and Reingold [37] algorithm was used to topographically place the nodes in the network: variables with many strong connections were placed in the center of the network and variables with weaker connections are placed more at the periphery of the network [37].

The same approach was applied to estimate and visualize the concentration network of associations between the five personality traits and the seven individual Insomnia Severity Index (ISI) item scores, providing a more detailed investigation of how personality traits are associated with different aspects of insomnia. We used polychoric correlations for the correlations involving ISI items to take the Likert-scale type variables into account.

To facilitate interpretation of the networks, we determined the “shortest paths” between all nodes. The shortest path between two nodes depends on the strength of the partial correlations between these two nodes, both directly—when present—and indirectly. The shortest path is defined as the “easiest” route through edges representing strong partial correlations [38]. As such, the shortest path between two nodes can be indirect, even when they are directly related. In addition, we determined centrality measures: strength, closeness, and betweenness [29]. The strength of a node is the sum of its absolute partial correlation coefficients. It summarizes the strength of the associations of the node with all its direct neighboring nodes. Closeness and betweenness in addition also take indirect relations of a node into account. The closeness of a node corresponds to the average “distance” between a node and all other nodes in the network [29]. The stronger the partial correlation between two nodes, the smaller their distance. Thus, the higher the closeness of a node, the stronger the average partial correlations to all other nodes. Betweenness represents the number of shortest paths that pass a node. The centrality measures give an idea of how strong the variance in a certain variable is associated with variance in the other variables. For example, if two persons have different scores on a very “central” variable, it is likely that their scores differ on most other variables as well. If these two persons have different scores on a variable that is less central, they may still have similar scores on the other variables.

## 3. Results

### 3.1. Data Description

Table 1 summarizes the measures of central tendency and dispersion for each of the variables. Table 2 provides the simple Pearson correlation coefficients among the personality traits and between each personality trait and the ISI sum score. All personality traits correlated significantly with each other, except for a lack of association of extraversion with conscientiousness, and of agreeableness with neuroticism. In agreement with previous reports, the simple Pearson correlations suggested that the ISI sum score was positively associated with neuroticism and agreeableness. In addition, unlike previous findings, openness was significantly associated with ISI sum score. In our sample, there was no significant correlation between extraversion or conscientiousness and ISI sum score.

Table 3 shows the polychoric correlation coefficients between the personality traits and the individual ISI items. Neuroticism, agreeableness, and openness were significantly associated with all ISI items. Extraversion and conscientiousness had significant associations with only some of the ISI items (Table 3), which corresponds to the absence of a significant association with the ISI sum score (Table 2).

### 3.2. Network Analysis of Personality Traits and ISI Sum Score

Figure 1 shows the concentration network of partial correlations of the five personality traits and the ISI sum score. Shortest paths are shown in solid lines, the rest in dashed lines. Neuroticism and ISI showed the strongest partial correlation, even stronger than the partial correlation between any two personality traits. This indicates that neuroticism has a stronger association with insomnia severity than with any other personality trait. Conscientiousness and extraversion were only indirectly related to insomnia via neuroticism. Neuroticism, extraversion, and agreeableness scored high on the three centrality measures (Figure 2), and neuroticism emerged as most central in the associations among personality traits and the ISI sum score.

### 3.3. Network Analysis of Personality Traits and ISI Items

Figure 3 shows the concentration network of partial correlations of the five personality traits and each of the seven ISI items. Shortest paths are shown in solid lines, the other paths in dashed lines. Most partial correlations among ISI items were much stronger than the partial correlations between the ISI items and the personality traits. The ISI items DIS, DMS, and EMA, which represent the nocturnal complaints of insomnia, clustered together. Likewise, the daytime complaints, items IDF, NIQoL, and Worry also clustered together. The shortest paths between the nocturnal complaints to the daytime complaints were all through their common associations with dissatisfaction with sleep. Nocturnal and daytime complaints thus emerge as two smaller clusters within the network of associations, and are bridged by Dissat, suggesting that nocturnal and daytime complaints separately feed dissatisfaction. Dissatisfaction thus gets a central role in the network of nocturnal and daytime insomnia characteristics measured by the ISI (Figure 4). Within the cluster of nocturnal complaints, DMS and DIS are most strongly associated with parts of the network representing personality trait association: DMS with conscientiousness and DIS with neuroticism. Within the cluster of daytime complaints, IDF is most strongly associated with parts of the network representing personality trait association through its associations with, once more, conscientiousness and neuroticism. A high betweenness, closeness, and strength for ISI items IDF and DMS (Figure 4) suggests that individuals that are more alike with respect to these two insomnia characteristics, are also more likely to resemble each other with respect to personality traits. Since ISI items DIS, EMA, and NIQoL scored low on all three centrality measures, such matching personality is less likely for subjects that resemble each other on these insomnia characteristics.

Whereas the shortest paths of the ISI sum score with personality traits involved neuroticism, agreeableness, and openness (Figure 1), a different picture emerged for the shortest paths between individual ISI items and personality traits. Neuroticism was the only trait that consistently connected the cluster of personality traits both with overall insomnia severity as well as with the cluster of individual insomnia complaints. However, the shortest paths between individual ISI items and personality traits now also included conscientiousness, rather than agreeableness and openness.

## 4. Discussion

The aim of this study was to obtain an integrated view on the associations between the five factor model personality traits and insomnia, both at the level of overall insomnia severity as well as at the level of individual symptom severities. In line with previous results, the simple Pearson correlations suggested that the personality traits neuroticism and agreeableness were positively related to insomnia severity [7,8,18,20,22,25,26]. Unlike previous studies, simple Pearson correlations also suggested a negative association between openness and insomnia severity, and no significant association between conscientiousness and insomnia severity. Most personality traits showed highly significant, small- to moderately-sized correlations with each other. This multi-collinearity makes it difficult to discriminate direct and indirect associations of insomnia severity with the individual personality traits. Therefore, we here applied a network approach to distinguish between direct and indirect associations.

We first estimated the network of partial correlations between personality traits and the overall insomnia severity as measured with the ISI summary score. Similar to the simple correlations, insomnia severity was most strongly and directly related to Neuroticism and secondarily as well to Openness and Agreeableness, and not directly related to Extraversion and Conscientiousness. In addition to the simple correlations, the network indicates possible pathways of influence that extent further than just two variables. For example, although extraversion is not directly related to insomnia, it is strongly associated to neuroticism and agreeableness, which in turn are related to insomnia. Thus, evaluating and visualizing the associations as a network provides insight into possible pathways across multiple variables.

We subsequently estimated a network of the partial correlations between personality traits and different insomnia complaints, as measured by individual ISI items. Evaluating the item-level networks provided a number of important insights, both on the association between insomnia complaints and personality and on the associations between insomnia complaints.

First, consistent with the previous analyses, neuroticism was directly related to insomnia complaints. Item-level analyses indicated that the strongest direct associations to personality concerned difficulty initiating sleep and interference with daily functioning. Interestingly, unlike the lack of a direct association of conscientiousness with overall insomnia severity, this personality trait did show direct associations specifically with the insomnia complaints of difficulty maintaining sleep and of interference with daily functioning. Notably, while the partial correlation between conscientiousness and interference with daily functioning was negative, the partial correlation between conscientiousness and difficulty maintaining sleep was positive. This suggests that while highly conscientious people are more likely to experience difficulty maintaining sleep, they are less likely to report that sleep problems interfere with their daily functioning. The inverse associations cancel out with the use of an overall insomnia severity measure, underscoring the value of item-level analyses.

Second, the network approach revealed that daytime and nocturnal insomnia complaints seem organized in two separable clusters, that both contributed to dissatisfaction. This indicates that a summary score may dilute possible specific daytime or nocturnal insomnia severity. The finding could moreover have consequences for the treatment of choice for insomnia, which is cognitive behavioral therapy (CBT-I) [39]. There could be additive effects of combining interventions that address nocturnal complaints with interventions that promote coping with daytime complaints. A second possibility is that interventions that specifically promote coping with dissatisfaction could ameliorate both daytime and nocturnal complaints. Indeed, CBT-I encompasses components focusing on managing expectations and beliefs regarding sleep, improving nocturnal sleep, and coping with daytime complaints [40].

The direction of influences between the insomnia complaints cannot be derived from the current cross-sectional assessment: future studies may consider repeated assessments, both during the development of insomnia as well as during intervention studies. By studying the effect of CBT-I on the individual insomnia complaints, strengths, and weaknesses of the intervention could be identified and efficacy may be improved.

Both network analyses showed that neuroticism is strongly and directly related to insomnia. This is in line with previous literature [7,8,18,20,22,25,26]. Assessing neuroticism may allow for early detection of premorbid predisposition of insomnia, the first of Spielman’s 3 Ps [5]. Coping with neuroticism has been shown feasible using both cognitive and cognitive behavioral interventions [41,42,43]. Moreover, knowledge about the personality traits that are characteristic of insomnia may provide clues on underlying causes of vulnerability to develop insomnia. For example, individual differences in personality traits are associated with individual differences in brain structure and brain function [12,13,14].

A few limitations of this study should be mentioned. First of all, participants filled out the questionnaires in an uncontrolled setting and at a self-chosen time. This resulted in a median of 7.4 months between filling out the ISI and the Mini-IPIP. However, since the Mini-IPIP assesses personality traits that are not assumed to change over time, the personality traits can be expected to be the same at the time the participants filled out the ISI. Another limitation, that has already been mentioned, is the cross-sectional set-up of this study. Whereas the current cross-sectional approach has revealed the strongest direct associations between constructs of personality and insomnia complaints, it requires longitudinal studies to address possible changes in the associations between ISI items. We recommend longitudinal and intervention studies to not only report on ISI summary scores, but also to investigate the network of associations between items.

## 5. Conclusions

In conclusion, using network analysis in a large sample, we obtained an integrated view on the associations of personality traits with both overall insomnia severity and individual insomnia complaints. We found that examining individual insomnia complaints provides additional information on the direct and indirect associations both between personality traits and insomnia, as well as between the different insomnia complaints. The approach allowed us to discriminate direct associations from indirect relations and thereby identify possible targets for improving CBT-I with the highest probability of effectively changing the network of associated complaints.

## Figures and Tables

**Figure 1 brainsci-07-00028-f001:**
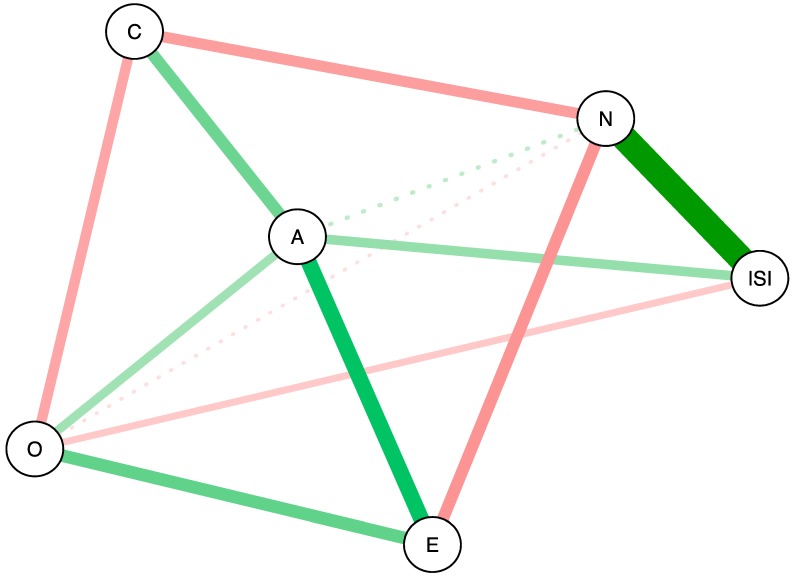
Concentration network of personality traits and the Insomnia Severity Index (ISI) sum score. Green lines indicate positive partial correlations; red lines indicate negative partial correlations. Solid lines indicate shortest paths, the other partial correlations are shown as dashed lines. The color saturation, thickness, and length of the edges represent the strength of the association. Abbreviations: A = agreeableness, C = conscientiousness, E = extraversion, N = neuroticism, O = openness. Neuroticism, agreeableness, and openness show direct relations to ISI, while conscientiousness and extraversion are indirectly related to ISI. The partial correlation between neuroticism and ISI is stronger than any of the other associations.

**Figure 2 brainsci-07-00028-f002:**
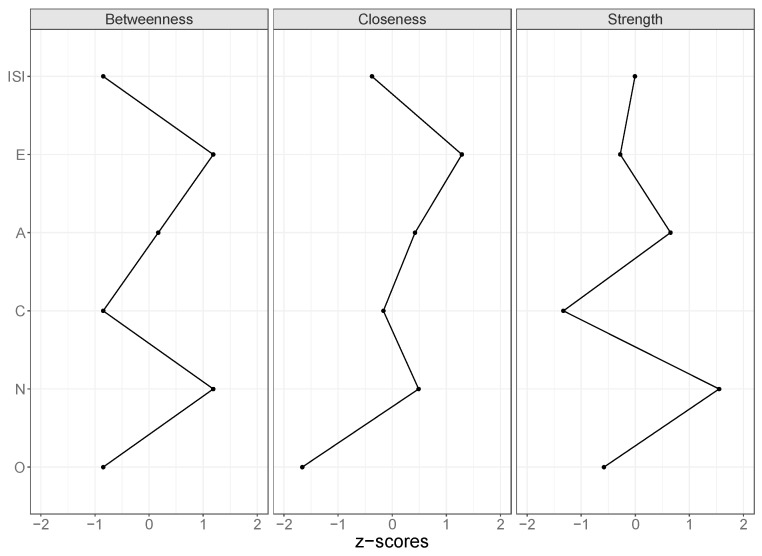
Standardized scores on the centrality measures of personality traits and the Insomnia Severity Index (ISI) sum score. Abbreviations: A = agreeableness, C = conscientiousness, E = extraversion, N = neuroticism, O = openness. The plots show that the personality traits neuroticism and extraversion are highest on all three centrality measures. Extraversion lies on the shortest path between two other nodes most often (**Betweenness**) and has the smallest overall distance to all other nodes (**Closeness**), while neuroticism is connected the strongest to its direct neighbors (**Strength**).

**Figure 3 brainsci-07-00028-f003:**
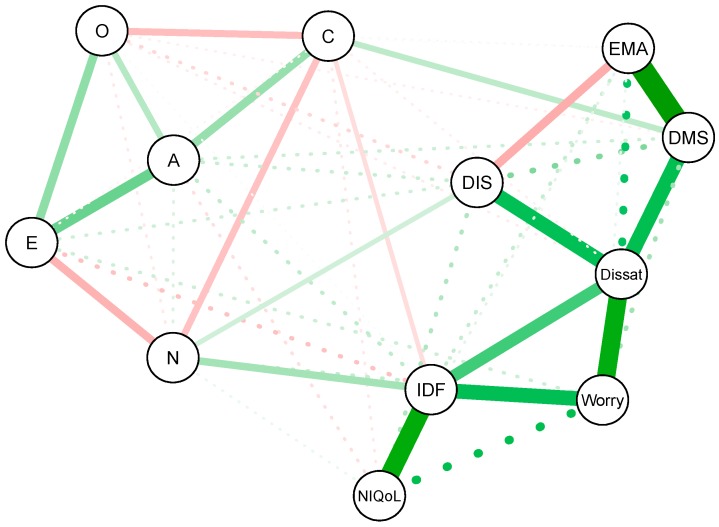
Concentration network of personality traits and Insomnia Severity Index (ISI) items. Green lines indicate positive partial correlations; red lines indicate negative partial correlations. Solid lines indicate shortest paths, the other partial correlations are shown as dashed lines. The color saturation, thickness and length of the edges represent the strength of the association. Abbreviations: DIS = Difficulty Initiating Sleep, Dissat = Dissatisfaction with sleep, DMS = Difficulty Maintaining Sleep, EMA = Early Morning Awakening, IDF = Interference with Daily Functioning, NIQoL = Noticeability of Impaired Quality of Life, Worry = Worry about sleep, A = Agreeableness, C = Conscientiousness, E = Extraversion, N = Neuroticism, O = Openness. The graph shows two main clusters of related variables, corresponding to a personality cluster and an insomnia cluster. The insomnia cluster can be further divided into a cluster of daytime symptoms (NIQoL, IDF, and Worry) and a cluster of nocturnal symptoms (DIS, DMS, EMA) that are connected via Dissat. The shortest paths that connect the personality and insomnia cluster contain neuroticism, conscientiousness, IDF, DMS, and DIS.

**Figure 4 brainsci-07-00028-f004:**
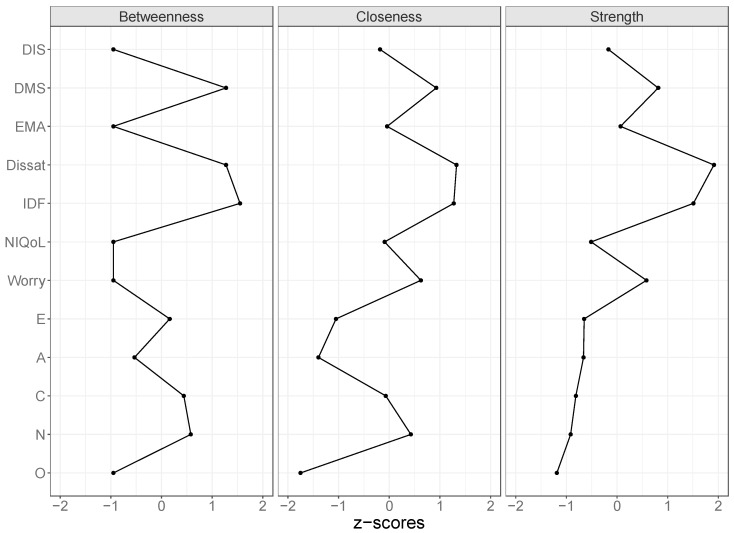
Standardized scores on the centrality measures of the personality traits and the ISI item scores. Abbreviations: Worry = Worry about sleep, NIQoL = Noticeability of Impaired Quality of Life, IDF = Interference with Daily Functioning, Dissat = Dissatisfaction with sleep, EMA = Early Morning Awakening, DMS = Difficulty Maintaining Sleep, DIS = Difficulty Initiating Sleep, A = Agreeableness, C = Conscientiousness, E = Extraversion, N = Neuroticism, O = Openness. The plots show that the ISI items Dissat and IDF are the highest on the centrality measures. IDF lies on the shortest path between two other nodes most often (**Betweenness**), while Dissat has the smallest overall distance to all other nodes (**Closeness**), and is connected the strongest to its direct neighbors (**Strength**).

**Table 1 brainsci-07-00028-t001:** Mean (M), standard deviation (SD) and observed range for the five personality traits, the Insomnia Severity Index (ISI) sum score and the separate ISI items.

	M ± SD	Range
Personality (IPIP-mini)		
Extraversion	12.45 ± 3.66	4–20
Agreeableness	16.94 ± 2.70	4–20
Conscientiousness	14.56 ± 3.43	4–20
Neuroticism	11.25 ± 4.03	4–20
Openness	14.89 ± 3.13	5–20
Insomnia (ISI)		
ISI sum	10.61 ± 7.20	0–28
DIS	1.19 ± 1.30	0–4
DMS	1.80 ± 1.49	0–4
EMA	1.42 ± 1.37	0–4
Dissat	2.10 ± 1.25	0–4
IDF	1.60 ± 1.29	0–4
NIQoL	1.14 ± 1.09	0–4
Worry	1.36 ± 1.25	0–4

Abbreviations: DIS = Difficulty Initiating Sleep, DMS = Difficulty Maintaining Sleep, EMA = Early Morning Awakening, Dissat = Dissatisfaction with sleep, IDF = Interference with Daily Functioning, NIQoL = Noticeability of Impaired Quality of Life, Worry = Worry about sleep.

**Table 2 brainsci-07-00028-t002:** Pearson correlation coefficients and corresponding *p*-values for the correlations among the five personality traits and between the five personality traits and the Insomnia Severity Index sum score.

	Extraversion		Agreeableness		Conscientiousness		Neuroticism		Openness	
	*r*	*p*	*r*	*p*	*r*	*p*	*r*	*p*	*r*	*p*
Extraversion										
Agreeableness	**0.24**	<10^−26^								
Conscientiousness	0.02	0.31	**0.15**	<10^−11^						
Neuroticism	**−0.17**	<10^−15^	0.03	0.13	**−0.15**	<10^−11^				
Openness	**0.20**	<10^−20^	**0.13**	<10^−8^	**−0.11**	<10^−6^	**−0.08**	<10^−4^		
ISI sum	−0.04	0.07	**0.12**	<10^−7^	−0.03	0.17	**0.38**	<10^−72^	**−0.10**	<10^−4^

Significant correlations are in shown in **bold** font.

**Table 3 brainsci-07-00028-t003:** Polychoric correlation coefficients and corresponding *p*-values for the correlations between each of the five personality traits and each of the individual Insomnia Severity Index items.

	Extraversion		Agreeableness		Conscientiousness		Neuroticism		Openness	
	*r*	*p*	*r*	*p*	*r*	*p*	*r*	*p*	*r*	*p*
DIS	0.01	0.80	**0.11**	<10^−7^	−0.02	0.30	**0.29**	<10^−40^	**−0.09**	<10^−4^
DMS	−0.02	0.34	**0.14**	<10^−10^	**0.06**	0.01	**0.28**	<10^−38^	**−0.10**	<10^−4^
EMA	−0.03	0.20	**0.09**	<10^−4^	0.03	0.18	**0.25**	<10^−30^	**−0.08**	<10^−3^
Dissat	**−0.04**	0.05	**0.11**	<10^−6^	−0.04	0.06	**0.36**	<10^−64^	**−0.10**	<10^−5^
IDF	**−0.09**	10^−4^	**0.11**	<10^−6^	**−0.10**	<10^−5^	**0.42**	<10^−89^	**−0.06**	0.01
NIQoL	**−0.05**	0.01	**0.06**	0.01	**−0.09**	<10^−4^	**0.34**	<10^−57^	**−0.06**	0.01
Worry	−0.03	0.20	**0.11**	<10^−6^	**−0.05**	0.03	**0.38**	<10^−71^	**−0.08**	10^−3^

Abbreviations: DIS = Difficulty Initiating Sleep, DMS = Difficulty Maintaining Sleep, EMA = Early Morning Awakening, Dissat = Dissatisfaction with sleep, IDF = Interference with Daily Functioning, NIQoL = Noticeability of Impaired Quality of Life, Worry = Worry about sleep. Significant correlations are shown in **bold** font.

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
