# Peer review of "Insomnia and Personality—A Network Approach"

_brainsci, 2017, doi:10.3390/brainsci7030028_

Round 1
Reviewer 1 Report
In my opinion a well written and laconic piece of work. The novelty of the approach made for very interesting reading, and the results compliment existing theories on personality and insomnia/ vulnerability to insomnia.
I have a couple of recommendations that I think would make for a more rounded and useful discussion:
I think the discussion could benefit from more teasing out of 'possible targets for interventions'. Does the paper advocate a role for acceptance therapy for example? How does the recommended targeting differ from already accepted CBTi which often trains on cognitive restructuring techniques and reframing of dissatisfaction?
The authors start by outlining the 3P model, and personality being one of the predisposing factors. How might this work inform theory around prevention of insomnia disorder and the useful of sleep education? My point here being really that is predisposition is a justification in the intro to the paper, i think it out to also be interpreted in the results via discussion.
The relationship between openness and ISI is reported as novel, so I think worth while discussing the implications for his more than is at present.
Minor point: Page 3/11 line 138. Word (strength?) missing.
Author Response
We would like to thank the reviewer for appreciating the paper and the novelty of the approach, as well as for the valuable suggestions for improvement. We have addressed all comments. For clarification, we have copied the original comments of the reviewer in bold font, our answers in regular font, and any text copied from the revised document in italics. Please note that we attached a PDF file of the revised manuscript to ensure the correctness of the references tot the line numbers.
I think the discussion could benefit from more teasing out of 'possible targets for interventions'. Does the paper advocate a role for acceptance therapy for example? How does the recommended targeting differ from already accepted CBTi which often trains on cognitive restructuring techniques and reframing of dissatisfaction?
We thank the reviewer for suggesting to make edits to our discussion. Both reviewers emphasized that the discussion could benefit from more elaborate in-depth explanations of the implications of our findings. For this particular point on possible targets of interventions, we have made changes to the discussion [lines 303-317]. We would like to emphasize that our results show the importance of taking the individual items into account, but that more, longitudinal, studies are necessary before changes to treatment can be suggested. Furthermore, we acknowledge that the ‘targets for intervention’ are already part(s) of CBT-I, but that its efficacy may be improved when knowing which insomnia complaints are affected most by it. We hope that we have clarified our recommendations.
The authors start by outlining the 3P model, and personality being one of the predisposing factors. How might this work inform theory around prevention of insomnia disorder and the useful of sleep education? My point here being really that if predisposition is a justification in the intro to the paper, i think it out to also be interpreted in the results via discussion.
We thank the reviewer for pointing out the important fact that we did not address parts of our introduction in the discussion. We have added a few lines to the discussion [lines 318-325], not only on how neuroticism as predisposing factor may play a role, but we also reflected on another remark we made in the introduction regarding personality traits and brain structure.
The relationship between openness and ISI is reported as novel, so I think worth while discussing the implications for his more than is at present.
We thank you for this suggestion. Although we indeed mention this relationship as different from previous findings, it only holds in our first network analysis. Given that it has not been reported before and we only find it in one of our analyses, we consider it too preliminary to really highlight this relationship. To take away the emphasis on Openness, we have added a remark about Conscientiousness, such that we discuss all findings in a similar way [lines 272-273].
Minor point: Page 3/11 line 138. Word (strength?) missing.
We thank the reviewer very much for pointing out this error, this was an erroneous sentence that should have been deleted before submission of the manuscript. We have taken it out.

Reviewer 2 Report
This is a new and interesting way of looking at insomnia and personality. It seems somewhat important to look at associations between the personality traits and the different questions of the Insomnia Severity Index. The network approach seems interesting, and the idea that this approach can give insight into which different aspects of personality can predict different aspects of insomnia relevant for interventions is important. However, I wonder if this can be done with the specific dataset, which is cross-sectional.
Some concerns:
Introduction: The introduction seems adequate and is well written. However: The authors state that: “there is no conclusive consistency about which of the personality traits are most strongly associated with insomnia” in line 50-51. This is partly true, but some of the research seems to provide more certain conclusions regarding this in specific populations, eg. the study of Vedaa et al. 2016, which is a prospective study of insomnia among nurses. I would therefor suggest that the authors add something like “in a general adult population” or “community sample”.
Also, the research by Hintsanen et al (2014) published in Health Psychology, may be relevant and should be considered included in the literature review.
Materials and methods: More information is needed concerning the procedure. When did the respondents answer the questionnaires? Did they answer them at home? Did they answer them several times? Also clearly state that it is a cross-sectional study.
Results: I am not familiar with the network approach, and it would be helpful if the authors could explain more clearly how this approach in this spesific study adds more to the understanding of personality and insomnia than the correlations.
Reporting tables and figures: I imagine this is the common, most established way to report network analysis results, however it is somewhat confusing for the reader. There are many tables and figures showing the same thing. Can these be collapsed? Are all of them needed? For example, could table 1 be included in table 2 and 3. What more do the figures really add that the tables do not tell us? Do we need both table 2 and figure 1? Also the explanation of Figure 1 is very similar to the explanation in the text.
Also; is it possible to add some information on which of the associations that are significant in the figures? It would also help if the authors could provide a reference to someone else who have reported the tables, figures ect. in the same way.
Discussion: The discussion seems more like a summary of the results, instead of a real in depth discussion about the importance of the findings in regards to earlier research and future research.
The findings on the correlations of the traits and the items of insomnia are interesting, and also between the different items of insomnia. But this is just correlations, which makes the results less strong. This is acknowledged in one sentence towards the end of the discussion (line 294-297), but should be problematized more. Also, more information on limitations of the study should be included.
The summary paragraph would be improved if it included a summary on what specifically was found – what is the most important finding from the study, what is the most valuable contribution from this paper? Also if targets for interventions should be mentioned it would be better if the authors explained which interventions they talk about.
Details:
P2, line 90, add % of females.
P 3, line 89- remove word “large” as it is relative what is a large or small n. The number speak for itself.
Author Response
We would like to thank the reviewer for appreciating the novelty and importance of our approach, and the helpful and thoughtful recommendations. We have addressed all comments. For clarification, we have copied the original comments of the reviewer in bold font, our answers in regular font, and any text copied from the revised document in italics. Please note that we attached a PDF file of the revised manuscript to ensure the correctness of the references tot the line numbers.
This is a new and interesting way of looking at insomnia and personality. It seems somewhat important to look at associations between the personality traits and the different questions of the Insomnia Severity Index. The network approach seems interesting, and the idea that this approach can give insight into which different aspects of personality can predict different aspects of insomnia relevant for interventions is important. However, I wonder if this can be done with the specific dataset, which is cross-sectional.
We thank the reviewer for the compliments and share the thought that longitudinal studies should be the next step. However, cross-sectional (partial) correlations are the first indication that certain direct associations are present in the construct and are a first indicator that associations between both personality and insomnia, as well as the different complaints of insomnia should be examined further. We added this suggestion to the discussion [lines 350-355].
The introduction seems adequate and is well written. However: The authors state that: “there is no conclusive consistency about which of the personality traits are most strongly associated with insomnia” in line 50-51. This is partly true, but some of the research seems to provide more certain conclusions regarding this in specific populations, eg. the study of Vedaa et al. 2016, which is a prospective study of insomnia among nurses. I would therefor suggest that the authors add something like “in a general adult population” or “community sample”.
We thank the reviewer for this suggestion of being more precise in our introduction and have added the proposed ‘in a general adult population’ in line 52.
Also, the research by Hintsanen et al (2014) published in Health Psychology, may be relevant and should be considered included in the literature review.
We thank the reviewer for suggesting to include this article. Without elaborating on it specifically, its contents were already summarized, together with other previous work, in the introduction (reference 20, lines 50-59).
Materials and methods: More information is needed concerning the procedure. When did the respondents answer the questionnaires? Did they answer them at home? Did they answer them several times? Also, clearly state that it is a cross-sectional study.
We thank the reviewer for suggesting to elaborate on our sample and procedure. We have added information on how and where participants fill out the questionnaires [line 86-87], which can be done at home or work or on any other computer connected to the internet, at a self-chosen time. They can fill out the questionnaires once. Also, we added the term ‘cross-sectional’ to line 91.
Results: I am not familiar with the network approach, and it would be helpful if the authors could explain more clearly how this approach in this specific study adds more to the understanding of personality and insomnia than the correlations.
We thank the reviewer for the important question to clarify the additional value of the network analysis approach compared to simple correlations. An important difference is that networks systematically show the partial correlations between variables. Unlike simple correlations, the partial correlations represent the association between two variables after conditioning on all other variables. As a result, the associations represent unique and direct relations between variables. For example, the lack of a direct association between two variables suggests that it is unlikely that they can causally influence each other in a direct manner. In contrast, a direct association opens up the possibility of a causal pathway between the two variables (Epskamp, Network Psychometrics. Unpublished dissertation[1], section 2.2., p.15). In sum, network analysis adds to the understanding of personality and insomnia by revealing the most robust direct relations among constructs, which may not stand out as clearly in matrices of simple correlations. The revised version now elaborates more on the added value of network analysis by defining partial correlations in the statistical analyses [lines 116-118], by explaining how the network associations can be interpreted [lines 119-122], and by interpreting the added understanding of the network analysis compared to correlations more in the discussion [lines 281-286]. In addition, to be as precise as possible, we mentioned the use polychoric correlations rather than Pearson or Spearman correlation to account for non-normality of the Likert-scale type ISI items [line 139-140]. Upon checking our correlations we detected that, although we used the polychoric partial correlations in our network analysis, we reported the Pearson correlations (table 3). We corrected this error [line 188]
Reporting tables and figures: I imagine this is the common, most established way to report network analysis results, however it is somewhat confusing for the reader. There are many tables and figures showing the same thing. Can these be collapsed? Are all of them needed? For example, could table 1 be included in table 2 and 3. What more do the figures really add that the tables do not tell us? Do we need both table 2 and figure 1? Also the explanation of Figure 1 is very similar to the explanation in the text.
An important difference between the tables and figures is that the tables give the correlations between the variables, while the figures shows the partial correlation network of the variables. We have tried to clarify this by stating more clearly that the connecting lines (edges) in the network graph represent the partial correlations [lines 128-129]. We thank the reviewer for mentioning the overlap in the explanation of Figure 1 and adjusted the explanation of Figure 1 in the text [lines 194-195]. Although we contemplated merging some of the tables, in the end we felt it would not facilitate the interpretation of the results and, in our opinion, make them complex.
Also; is it possible to add some information on which of the associations that are significant in the figures? It would also help if the authors could provide a reference to someone else who have reported the tables, figures etc. in the same way.
We thank the reviewer for the important question to clarify the interpretation of the networks in terms of significance. The edges in the network are not individually tested for significance. Instead, we applied a regularization technique called the least absolute shrinkage and selection operator (LASSO), which is commonly used in network analyses to account for spurious associations (Epskamp, Network Psychometrics. Unpublished dissertation[1], section 2.3., p.16). The reason to use regularization instead of significance testing is as follows. When all associations must be tested for significance, many statistical tests have to be performed ((n*(n-1))/2). Controlling for multiple testing would result in a loss of power. Regularization, on the other hand, can account for spurious associations by shrinking small associations to zero, by which it selects only the most robust associations and minimizes the false positive associations. This way, the associations that are retrieved by the LASSO are very likely to play an important role in the network. We tried to explain the role of the LASSO regularization better in the statistical analyses [lines 123-125] and provide a reference to a tutorial that explains how to conduct and report network analyses (reference 29).
Discussion: The discussion seems more like a summary of the results, instead of a real in depth discussion about the importance of the findings in regards to earlier research and future research.
We thank the reviewer for suggesting to make edits to our discussion. Both reviewers emphasized that the discussion could benefit from more elaborate in-depth explanations of the implications of our findings. We have added a section in which we reflect on some remarks we made in the introduction, namely the 3P model and the link between personality traits and brain structure [lines 318-325]. Also, we changed the paragraph that discussed possible improvement of interventions for insomnia, emphasizing that our cross-sectional study doesn’t provide conclusive evidence, but is an important first step in identifying possible targets for improvement [lines 303-317].
The findings on the correlations of the traits and the items of insomnia are interesting, and also between the different items of insomnia. But this is just correlations, which makes the results less strong. This is acknowledged in one sentence towards the end of the discussion (line 294-297), but should be problematized more. Also, more information on limitations of the study should be included.
We thank the reviewer for finding our approach interesting and, as mentioned above, we share the thought that cross-sectional studies should be supplemented with longitudinal data in order to find conclusive evidence for targets of interventions. Cross-sectional (partial) correlations are the first indication that certain direct associations are present in the construct and are an important first indicator that both the associations between personality and insomnia, but also the different complaints of insomnia should be examined further. We have added a paragraph to the discussion explaining this, and in addition mentioned a few other limitations of the study [line 326-355].
The summary paragraph would be improved if it included a summary on what specifically was found – what is the most important finding from the study, what is the most valuable contribution from this paper? Also if targets for interventions should be mentioned it would be better if the authors explained which interventions they talk about.
We thank the reviewer for this suggestion and have added more information on the most important findings. Also, we have named the interventions we mentioned [lines 358-361].
Details:
P2, line 90, add % of females.
P 3, line 89- remove word “large” as it is relative what is a large or small n. The number speak for itself.
We thank the reviewer for suggesting both changes. We have made the necessary edits.
[1] Which can be accessed through: http://sachaepskamp.com/Dissertation

Round 2
Reviewer 2 Report
The authors have answered all of my queries sufficiently.